# Potential Roles of O-GlcNAcylation in Primary Cilia- Mediated Energy Metabolism

**DOI:** 10.3390/biom10111504

**Published:** 2020-11-01

**Authors:** Jie L. Tian, Farzad Islami Gomeshtapeh

**Affiliations:** 1Complex Carbohydrate Research Center, Department of Biochemistry and Molecular Biology, University of Georgia, Athens, GA 30602, USA; 2Department of Biochemistry and Molecular Biology, University of Georgia, Athens, GA 30602, USA; fi52622@uga.edu

**Keywords:** primary cilia, O-GlcNAc, energy homeostasis, obesity, diabetes

## Abstract

The primary cilium, an antenna-like structure on most eukaryotic cells, functions in transducing extracellular signals into intracellular responses via the receptors and ion channels distributed along it membrane. Dysfunction of this organelle causes an array of human diseases, known as ciliopathies, that often feature obesity and diabetes; this indicates the primary cilia’s active role in energy metabolism, which it controls mainly through hypothalamic neurons, preadipocytes, and pancreatic β-cells. The nutrient sensor, O-GlcNAc, is widely involved in the regulation of energy homeostasis. Not only does O-GlcNAc regulate ciliary length, but it also modifies many components of cilia-mediated metabolic signaling pathways. Therefore, it is likely that O-GlcNAcylation (OGN) plays an important role in regulating energy homeostasis in primary cilia. Abnormal OGN, as seen in cases of obesity and diabetes, may play an important role in primary cilia dysfunction mediated by these pathologies.

## 1. Introduction

### 1.1. Primary Cilia in Regulating Energy Homeostasis

Primary cilia were first discovered in 1898 and given this name by Sergei Sorokin in 1968 [1,2]. They were considered vestigial organelles until recent research revealed their presence on most types of quiescent mammalian cells, including neurons, pancreatic endocrine cells, respiratory tract cells, and smooth muscles [2,3,4,5]. Primary cilia are composed of a nine-doublet microtubule (DMT)-based axoneme enclosed by a highly modified ciliary membrane [6]. Transportation of ciliary proteins is carried out by intraflagellar transport (IFT), which carries axoneme proteins and cargos from the cytoplasm to the ciliary compartment through movement along the DMTs [7]. Recent findings indicate that the BBSome complex also participates in the transportation of ciliary membrane proteins as a subset of the IFT complex [8,9]. The BBSome core complex is composed of at least seven Bardet-Biedl syndrome (BBS) proteins [10]. While some BBS mutations result in assembly of cilia with only minor functional defects, BBS loss-of-function mutations, such as *bbs-1, 7*, and *8*, result in dissociation of the IFT complex and complete ciliary dysfunction [11,12,13]. Primary cilia on different tissue types are concentrated with various receptors, ion channels, and transporters to transduce extracellular signals within the cell and to play roles in a variety of key processes, including development, organ function, and tissue homeostasis [3,14,15]. Irregular formation or dysfunction of primary cilia cause a number of human diseases, such as retinal degeneration, polycystic kidney disease, the Bardet-Biedl syndrome, and the Joubert syndrome, collectively known as ciliopathies [16]. Ciliopathies usually coincide with obesity, diabetes, metabolic disorder, cancer, and neurodegenerative diseases, suggesting that the functions of primary cilia include metabolic regulation and maintenance of energy homeostasis [17]. Dysregulation of energy metabolism is a major cause of obesity [18]; obesity is furthermore associated with insulin resistance and type 2 diabetes mellitus (T2DM) [19,20]. Therefore, understanding the pathogenesis and molecular mechanisms of obesity and diabetes involved in ciliary defects is important in developing therapeutic paradigms [17]. Functions of primary cilia in regulating energy homeostasis have been widely studied in the mouse model system as well as in humans and are described in the following three aspects.

#### 1.1.1. Hypothalamic Neuronal Cilia Regulate Energy Balance

Recent molecular studies reveal that the hypothalamus of the central nervous system (CNS) is the central hub that balances feeding, energy expenditure, and regulates the energy homeostasis in mammals [21,22]. The hypothalamus is composed of distinct types of nuclei that respond by secreting neuronal hormones into the bloodstream when receiving peripheral hormones without causing blood-brain barrier leakage. The peripheral hormones may include growth hormone, prolactin, insulin, insulin-like growth factor- I/II, and leptin. The arcuate nucleus (ARC) of the hypothalamus is critical in sensing and responding to the peripheral metabolic hormones and nutrients signals. The ARC is adjacent to the median eminence (ME) and third ventricle, where it has easy access to such signals [23,24]. ARC neurons are ciliated, and the neuronal cilia are major players in sensing metabolic signals and maintaining energy homeostasis [25,26]. Conditional disruption of neuronal cilia, either throughout the central nervous system or from pro-opiomelanocortin (POMC)-expressing cells in the hypothalamus, results in hyperphagic-related obesity in mice and is accompanied by elevated levels of serum insulin, glucose, and leptin [26]. Consistent with Davenport’s results, artificially shortening hypothalamic neuronal cilia length by injecting siRNA specific to ciliogenic genes *Kif3a* or *Ift88* in mice resulted in increased food intake, dampened energy expenditure, and weight gain. In addition, shortened cilia attenuated anorectic responses to insulin, glucose, and leptin [27]. These findings indicate that neuronal cilia are critical for sensing metabolic signals, such as hormonal and nutritional levels, and that cilia-mediated signaling functions in controlling energy balance.

#### 1.1.2. Primary Cilia Control Energy Homeostasis and Inhibit Adiposeness 

Adipocytes in adipose tissues are another critical variable in the control of energy balance since they regulate lipid storage and hormone secretion of leptin and cytokines [17,28]. Adipocytes are differentiated from the preadipocyte lineage derived from Adipose-derived mesenchymal stem cells (ASCs), which possess differentiation capability in response to adipogenesis and angiogenesis [29,30]. Adipocytes secrete the hormone leptin to decrease appetite, reduce plasma insulin levels, and increase metabolite rate [31]. The secretion of leptin from adipocytes is stimulated by insulin secretion, glucose uptake, and nutrient availability, which are all factors that promote anabolism [32,33]. Multiple lines of evidence have shown that cilia-related signaling is involved in controlling adipogenesis. Primary cilia are present in the preadipocytes during differentiation, which is the transition period of proliferating preadipocytes to matured adipocytes [34]. The cilia, during this transition period, carry Wnt and Hedgehog (HH) receptors that potentially act as adipogenesis inhibitors and maintain the undifferentiated status of preadipocytes [12,35]. Impaired cilium formation in preadipocytes favors adipogenesis and is accompanied by increased fat levels and higher leptin secretion by adipocytes [34]. Moreover, ASCs from obese donors (BMI [body mass index] >35) have shortened primary cilia compared to control donors (BMI <25) [36]. In vivo and in vitro data indicate that primary cilium dysfunction may promote adipogenesis, thus contributing to the development of dysregulation of energy homeostasis, leptin resistance, and obesity.

#### 1.1.3. Primary Cilia Regulate Pancreatic β-Cell Insulin Secretion to Balance Energy Metabolism

Findings showed that dysfunction or decreased expression of ciliary genes impairs beta-cell proliferation and insulin secretion in the pancreas and promotes the development of type 2 diabetes in rodents and humans. Besides, almost two-thirds of genes involved in T2DM susceptibility are found in the ciliary proteome database [37,38]. Primary cilia are present in the α, β, and δ cells of the Langerhans islet of the endocrine pancreas [39,40]. Insulin receptors (IRs) localize to the primary cilia of stimulated β-cells, and the activation of downstream insulin signaling requires integrity of the ciliary/basal body. Knockdown of the basal body genes *Oral-facial-digital syndrome I* (*Ofd1*) and *Bbs4* impairs insulin secretion, causing defects in the ciliary localization of IRs, and attenuating IR-mediated signaling activities, such as insulin receptor substrate proteins (IRS), phosphatidylinositol-3-kinase (PI3K), and serine/threonine protein kinase Akt/PKB (protein kinase B). Conversely, in vivo results showed that diabetic Goto-Kakizaki (GK) rats have significantly less ciliated β-cells compared to Wistar controls [37]. Overall, these findings suggest that there is a correlation between primary ciliary dysfunction and decreased β-cells secretion of insulin in the contribution of T2DM.

### 1.2. O-GlcNAc Functions in Maintaining Energy Homeostasis

O-linked β-N-acetylglucosamine (O-GlcNAc) modification was unexpectedly discovered in 1983 when bovine milk galactosyltransferase and UDP-[3H]galactose were used to probe murine immune cells for terminal N-acetylglucosamine residues [41]. The O-GlcNAc modification generally happens on over four thousand nuclear, mitochondrial, and cytoplasmic proteins of the cell [42,43]. A single pair of enzymes O-GlcNAc transferase (OGT) and O-GlcNAcase (OGA) regulate the dynamic cycling of O-GlcNAc on and off serine/threonine residues of target proteins. OGT uses UDP-GlcNAc, which is derived from the hexosamine biosynthetic pathway (HBP), as a substrate donor to glycosylate proteins. The production of UPD-GlcNAc by the HBP integrates with the metabolism of carbohydrates, amino acids, fats, and nucleotides, which renders the concentration of UDP-GlcNAc highly dependent on the nutrient flux and metabolite availability [44,45,46]. Moreover, the activity of OGT is highly responsive to UDP-GlcNAc concentrations. Therefore, the cellular OGN levels depend on the metabolite state of the cell and O-GlcNAcylation serves as a nutrient sensor [47,48]. Maintaining homeostasis of OGN is important for the normal function of many cellular processes, such as transcription, translation, and signal transduction [41]. Deregulation of OGN homeostasis contributes to the pathogenesis of a plethora of human diseases, including diabetes, neuro-degeneration, and cancer [49,50].

### 1.3. The Nutrient Sensor O-GlcNAc Contributes to Primary Ciliary Length Regulation 

Recent publications have documented the relationship between the regulation of cellular OGN levels and primary ciliary length. First, primary ciliary length negatively correlates with the cellular OGN levels both in hTERT-RPE1 and IMCD3 cells. Molecular mechanistic studies showed that OGN of α-tubulin and histone deacetylase 6 (HDAC6) are involved in ciliary shortening through promotion of axonemal microtubule disassembly [51]. Consistent with the work of Yu et al., primary cilia were fewer and shorter in the diabetic mouse eyes, trachea, and skin cells compared to those of wild type, and increased OGN of RPE1 cells in culture impaired ciliogenesis [52]. However, Yu’s results showed that genetic knockdown of OGT expression and pharmacological inhibition of OGT activity with benzoxazolinone core (BZX) also attenuated ciliary length. This partially agrees with the previous finding that glucose deprivation decreased ciliary length but promoted primary cilia formation [53]. The differing observations may be due to different cell culture status and treatments. Besides, obese individuals in humans also have shortened and compromised functional ASC primary cilia. Elevated Aurora A and HDAC6 activities contribute to the shortening of ciliary length, and deficient HH signaling may promote ASCs to differentiate into adipocytes in obese individuals [36]. Considering escalated OGN levels are tightly correlated with diabetes and glucose availability [54,55], these findings suggest a potential role of O-GlcNAc in primary ciliary length regulation. Intriguingly, it has been noticed that many components of cilia-mediated metabolic signaling pathways are either O-GlcNAcylated or have O-GlcNAc-regulated activity. We hypothesize that the cross-regulation between O-GlcNAc and primary cilium is important in maintaining energy homeostasis and that O-GlcNAc dysregulation contributes to primary ciliary dysfunction associated with the pathogenesis of obesity and diabetes.

## 2. Main Text

### 2.1. Interplay between Neuronal Primary Cilia and O-GlcNAnac in Regulating Energy Homeostasis in Hypothalamus

#### 2.1.1. Hypothalamic Neurons Regulate Energy Homeostasis

The ARC in the hypothalamus plays a pivotal role in response to metabolic hormones and nutrients signals. The ARC is composed of two groups of neurons that transduce orexigenic (appetite stimulating) and anorexigenic (appetite inhibiting) stimuli, respectively. One group produces neuropeptide Y (NPY) and agouti-related peptide (AgRP), while the other produces pro-opio-melanocortin (POMC) and cocaine- and amphetamine-regulated transcript (CART). These two neuron groups communicate with each other through various hormone receptors distributed over the neuronal membrane in order to mediate signal transduction and to control food intake and energy homeostasis; such receptors include leptin receptor (LepR), insulin receptor (IR), melanocortin 4 receptor (MC4R), neuropeptide Y receptors Y1 and Y2 [56,57]. Leptin inhibits AgRP/NPY neurons and activates POMC neurons to discourage feeding by binding to the leptin receptor (LepR) on the plasma membrane. Upon leptin-LepR binding, the expression of neuropeptides NPY and AgRP is reduced. The expression of α-melanocyte-stimulating hormone (α-MSH), however, is induced from POMC [27,58]. AgRP and α-MSH are ligands acting antagonistically on the same melanocortin 3 and 4 receptors (MC3/4R) on the lateral ARC neurons in the ventral posterior nucleus (VPN) of the thalamus. α-MSH activates MC4R to reduce food intake, while AgRP inhibits MC4R to increase feeding [59,60]. Please refer to the review paper by Spiegelman et al. for more detailed descriptions of leptin-regulated energy balance in central neural circuits [19]. Signaling pathways through insulin receptor (IR) on POMC and AgRP/NPY neurons are another major factor in regulating glucose and energy homeostasis [61,62]. Insulin activates POMC neurons to increase metabolism while suppressing the fire rate of AgRP/NPY neurons and reducing gluconeogenesis in the liver to decrease energy metabolism [61,63]. Leptin and insulin are interconnected to regulate food intake and energy homeostasis by acting on neuronal populations in the hypothalamic ARC to modulate output. For instance, insulin induces leptin secretion by adipose tissue and the two hormones work together to fire POMC neurons to accelerate energy expenditure by promoting white adipose tissue browning and weight loss [63,64].

#### 2.1.2. LepR- and IR-Mediated Metabolic Signaling Pathways Control Feeding, Energy and Glucose Homeostasis in the Hypothalamic Neurons 

The molecular mechanisms of LepR-mediated signaling pathways are well studied. The representative signaling pathway of LepR on the POMC is integrated with Janus kinase (JAK) and signal transducer and activator of transcription (STAT3) to regulate subsequent transcription of target genes involved in energy homeostasis, such as *POMC* and suppressor of cytokine signaling 3 (*SOCS3)* [65]. Activation of STAT3 through phosphorylation promotes POMC expression by binding to the *pomc* promoter, while the binding of STAT3 to the *agrp* promoter inhibits AgRP expression [66]. POMC expression suppresses hunger, while the induction of SOCS3 in turn inhibits both leptin and insulin pathways, making it a negative feedback on leptin-JAK2-STAT3 signaling [67]. The other most studied targets of LepR are the proteins phosphatidylinositol-3-kinase (PI3K), 3-phosphoinositide-dependent protein kinase 1 (PKD1), and protein kinase B (AKT) [68,69]. Leptin-mediated PI3K-PDK1-AKT signaling is activated both in POMC and AgRP neurons but with different effects on the lateral neurons. Leptin stimulates POMC neurons and inhibits AgRP neurons to reduce food intake [70,71].

Interestingly, the insulin and leptin signaling pathways are integrated through PI3K in ARC neurons [72]. The activity of hypothalamic neurons is modulated by the activation of PI3K-PDK1-AKT downstream proteins and transcription factors, including AKT actives AMP-dependent kinase (AMPK), Forkhead box protein O1 (FoxO1), and mammalian target of rapamycin (mTOR) [69,73]. Insulin signaling is mediated by insulin receptors (IRs) expressed on hypothalamic neurons POMC, AgRP, and NPY [61,74]. Deletion of IR on POMC or AgRP results in suppression of hepatic glucose production but shows no interference in energy homeostasis and no effect on satiety, indicating that the major function of insulin action is maintaining glucose homeostasis in hypothalamic neurons [59]. PI3K-PKD1-FoxO1 is one of the signaling pathways that modulates energy homeostasis and mediates leptin and insulin action in POMC and AgRP neurons. FoxO1, when unphosphorylated, is in its active form and resides in the nucleus to suppress the expression of *pomc* and stimulate the expression of *agrp* [75,76]. Activation of leptin and insulin leads to phosphorylation of FoxO1 and subsequent exportation from the nucleus. FoxO1 export leads to STAT3 binding to the *pomc* and *agrp* promoters because of their overlapping promoter-binding sites, which further results in the activation of POMC neuron and inhibition of AgRP neurons [77,78] (Figure 1). mTOR-AMPK is another mediator that participates in the leptin and insulin signaling pathways. When there is an energy surplus, leptin signaling activates mTOR, promoting phosphorylation and inhibition of AMPK to suppress food intake [20,77].

#### 2.1.3. Primary Cilia on the Hypothalamic Neurons are Critical for the Regulation of Energy Homeostasis

The function of primary cilia in the regulation of energy homeostasis is revealed in the mutation of BBS-associated genes in mice and humans, which ultimately leads to obesity in adulthood [11,79]. Davenport et al. first reveal the importance of primary cilia in the hypothalamus. The disruption of primary cilia in POMC or AgRP neurons through conditional knockout of ciliogenic genes *IFT88* and *Kif3a* promote hyperphagic induced obesity. Mice lacking cilia in adulthood become obese and display elevated serum leptin, insulin, and glucose levels [26]. These results indicate that the role of primary cilia on neurons in the hypothalamus in response to satiety signals, such as leptin and insulin, is to regulate energy homeostasis [26,61,71].

The possibility of LepR-b trafficking to the neuronal cilia basal body was proposed in several studies soon after the discovery that somatostatin receptor 3 (SSTR3) and melanin-concentrating hormone receptor 1 (MHCR1) target of the hypothalamic neuronal cilium. In the ciliary genes *BBS* knockout mice or retinitis pigmentosa GTPase regulator-interacting protein-1 (RPGRIP1L) haplo-insufficient mice, the trafficking of LepR to the periciliary area is attenuated, and the LepR signaling in hypothalamic neurons is disrupted [80,81,82]. Further studies demonstrated the importance of neuronal cilia by showing that disruption, specifically in the POMC neurons, significantly reduces leptin sensitivity [26,27]. The attenuated leptin sensation due to primary cilia dysfunction leads to increased food intake and body weight. Conversely, shorter ciliary length is observed in leptin-deficient *ob/ob* obese mice and LepR-deficient *db/db* diabetic mice, and this shortening of ciliary length is rescued by leptin treatment. Moreover, leptin elongates the cilia of hypothalamic neurons in vitro [25,27]. These data indicate a substantial correlation between primary ciliary length and energy metabolic status. Dysfunction of primary cilia disrupts energy homeostasis via mislocalization of hormone receptors. Conversely, deficiency in hormone or hormone receptors cause extra energy intake, hence improves glucose homeostasis and shortened cilia length.

#### 2.1.4. O-GlcNAc Regulates Energy Homeostasis in the Hypothalamus

The importance of O-GlcNAc in regulating energy homeostasis through hypothalamic neurons is reflected in the ablation of *OGT* in AgRP neurons. Once the AgRP neurons are activated by signals such as those caused by fasting, the browning of the white adipose tissue will be suppressed to inhibit energy expenditure. However, abolishing OGT in AgRP neurons blocks excitation of AgRP neurons, promoting browning of white adipose tissue and thus protecting the mutated mice against diet-induced obesity and insulin resistance [83]. Even though OGT expression and, therefore, OGN levels are higher in the hypothalamic neurons compared to the peripheral tissues, elevated O-GlcNAc levels act as a signal to activate AgRP neurons under fasting conditions [83]. Results indicate that O-GlcNAc-mediated signaling is critical in regulating energy homeostasis in the hypothalamic neurons.

Multiple lines of evidence have shown that O-GlcNAc is intensively cross-regulated with the insulin metabolic signaling pathway. Once insulin or insulin-like growth factors (IGF) bind to IRs, phosphorylation of adaptor signaling proteins insulin receptor substrate 1 or 2 (IRS-1/2) will trigger to catalyze the activation of downstream cascades PI3K, PDK1, and AKT [84]. Activation of AKT phosphorylates multiple downstream effectors, such as FoxO1, mTOR, AMPK, glycogen synthase kinase 3 β (GSK3β), and ribosomal protein S6 kinase (p70S6K) [85,86]. O-GlcNAc glycosylation occurs on most of the insulin signaling proteins listed above and is involved in regulating activity of proteins, including IRS-1, PI3K, PDK1, AKT, AMPK, FoxO1, GSK3β, and p70S6K [84,87] (Figure 1). OGN attenuates IRS-1 and AKT activity either by disrupting their interaction with PI3K and PDK1 kinase respectively or by interplaying with activated phosphorylated forms [49,88,89]. Muscle-specific OGT knockout mice showed higher energy expenditure, insulin signaling, and whole-body insulin sensitivity. Therefore, O-GlcNAc is considered a negative regulator of the insulin signaling pathway [89,90] Even though there is presently no evidence showing that IR is located on neuronal cilia directly, the trafficking of IR to the cell membrane is mediated by BBS proteins, and IR is recruited to the primary cilia of the stimulated pancreas β-cells, these indicate the potential localization of IR in the periciliary area [37,91].

Unlike insulin, O-GlcNAc is a positive regulator of the leptin JAK-STAT signaling pathway. Increased OGN levels are associated with leptin-mediated phosphorylation and activation of pSTAT3 (Y705) in the liver and hypothalamus of rats, which freely access to a 30% sucrose solution [92,93]. A recent study has shown that STAT3 is O-GlcNAcylated on T717, implying the interplay with phosphorylation to regulate STAT3 activity [94]. Moreover, the cross-talk between OGN and Jak2-mediated tyrosine phosphorylation has been detected in the peptide microarray assay for OGT and tyrosine kinase [95]. The findings implicate that O-GlcNAc is an activator of the cilia-mediated leptin-LepR-JAK2-STAT3 pathway in the hypothalamus.

#### 2.1.5. The Potential of O-GlcNAc Involves in the Dysfunction of Primary Ciliary Induced Obesity/Diabetes

The previous discussion showed that primary cilia are critical for proper transduction of metabolic signals in hypothalamic neurons. Investigations recently showed primary ciliary length tightly correlates with signal transduction and energy status in the body. In chow-diet-fed lean mice, the primary ciliary length on the hypothalamic ARC neurons is about 5.5 ± 0.44 μm on average. However, the average ciliary length reduced by 40–60% in the hypothalamus of diet-induced obese (DIO) mice fed with a high-fat, high-sucrose diet, and mice are accompanied by leptin resistance. Furthermore, in rodents, the maternal high-fat diet impairs leptin signaling at the ARC level, which leads to attenuated leptin-induced appetite suppression [96]. These results increase the evidence that overfeeding decreases ciliary length. The shortened cilia deteriorate the hormonal responses that affect energy metabolism and eventually result in obesity [27]. One molecular mechanism implied in this finding is that the defection of primary cilia diminishes hypothalamus neurons LepR trafficking to or near the cilium, effectively halting the effects of leptin on satiety and so contributing to obesity [80,97]. Besides, defection of primary cilia in the *Kif3a* and *Ift88* knockdown mice resulted in lower responsiveness to leptin, which is demonstrated by lower phosphorylated STAT3 levels [26].

Current findings showed that the JAK2 and PI3K signaling pathways are involved in leptin-mediated cilia elongation. Pharmacological inhibition of JAK2 and PI3K or genetical knockdown of PI3K expression blocks leptin-induced primary cilia elongation [27]. Downstream PI3K targets, such as PTEN, AKT, and GSK3β, are also involved in ciliary length regulation. Knockdown of *Pten* and *Gsk3b* increases ciliary length, while overexpression of *Pten* and *Gsk3b* decreases ciliary length and blocks the effects of leptin on hypothalamic neuron cilia. Mechanism studies showed that leptin treatment elongates the ciliary length by inhibiting PTEN and GSK3β signaling and by increasing the expression of IFT proteins [25,50]. These results strongly suggest that primary ciliary length is critical in stimulating LepR-mediated leptin signaling pathways in hypothalamic neurons. It is not surprising that insulin treatment increases ciliary length of hypothalamic neurons by up to 25% since insulin signaling is integrated with leptin signaling via the PI3K signaling pathway [27]. The observation may implicate that insulin promotes ciliary growth in parallel with leptin.

A large number of studies showed that leptin-mediated signaling molecules are involved in ciliary length regulation. However, the mechanisms of how they regulate primary cilia growth are still largely unknown. Nutrient sensor O-GlcNAc is an emerging factor in regulating the ciliary length in the hypothalamus. The work of Dai et al. showed that conditional knockout neuronal OGT by crossing CaMKIIα-Cre^(+)^ and Ogt^loxp(+)/loxp(+)^ mice lead to reversible increases in food intake, body weight gain, insulin resistance, and serum leptin level, which are all prediabetic symptoms [98]. Mechanism studies revealed that decreased neuron cells and LepR in the hypothalamus contributes to the upregulation of appetite and insulin/leptin resistance [98]. Another finding revealed that mutation in LepR induced diabetic (DB) mice (Lepr^db^/Nju) showed ciliary defection and OGA level elevation [52]. These two investigations provide a new perspective suggesting that O-GlcNAc negatively regulates ciliary length through its integration with LepR signaling to maintain energy homeostasis in the hypothalamus.

### 2.2. Primary Cilia Inhibit Adipocyte Differentiation to Regulate Energy Homeostasis

Overeating, or excess energy intake will cause lipogenesis, or storage of excess energy in the form of triglyceride in the lipid droplets of adipocytes. The increased fat mass is manifested both in greater adipocyte size and elevated adipogenesis [99]. Adipogenesis is the process of preadipocyte cells differentiating into adipocytes. The mesenchymal precursors commit to preadipocyte lineage cells then differentiate into mature adipocytes [100]. The upregulation of BBS genes during the early phase of adipogenesis unveiled the critical function of primary cilia in controlling preadipocyte differentiation [101,102]. Primary cilia are only present on the differentiating preadipocytes and absent on the proliferating or mature adipocytes. Wnt and HH receptors on the transit cilium mediate signaling pathways to repress adipogenesis [12,34,35,103]. The stimulation of Wnt phosphorylates and inactivates GSK3 to suppress phosphorylation of β-catenin. The unphosphorylated active form of β-catenin translocates to the nucleus and initiates the transcription of Wnt signaling molecules, which repress preadipocyte differentiation [103,104]. In the absence of Wnt signaling, the active form of GSK3β promotes nuclear accumulation of proliferator-activated receptor-γ (PPARγ), another master regulator of adipogenesis, to stimulate adipocyte differentiation [105,106] (Figure 2). Impaired ciliogenesis by knockdown of the genes expressing BBS 10 and BBS12 proteins results in the activation of GSK3β, degradation of β-catenin, and accumulation of PPARγ in the nucleus to increase adipogenesis [34,104]. Furthermore, the loss of function of ciliary protein retinitis pigmentosa GTPase regulator-interacting protein 1 like (RPGRIP1L), which localizes to the primary cilium transition zone and is required for Hh signaling, contributes to the 3T3-L1 preadipocytes differentiating into mature adipocytes. This effect, however, is not observed in mature 3T3-L1 adipocytes, which indicates the specific function of primary cilia in adipogenesis [107,108]. In combination, these results reveal the important function of primary cilia in suppressing preadipocyte differentiation.

Several observations have shown that O-GlcNAc plays functions in adipocyte differentiation. In the study of Ishihara et al., the OGN levels significantly increase in a differentiation-dependent manner in the mouse 3T3-L1 preadipocytes [109]. It takes 8 days for 3T3-L1 cells to differentiate from preadipocytes to matured adipocytes. The OGN levels increase drastically after the fifth day, simultaneously with the induction of transcription factor C/EBPα, which is critical for adipocyte differentiation. Pharmacological inhibition of HBP or O-GlcNAc cycling blocks the progress of differentiation and C/EBPα expression [109,110]. Results indicate that increased OGN levels function in the induction of adipocyte differentiation. The latter study showed that PPARγ is O-GlcNAcylated and that the O-GlcNAc modification is essential for the stability and transcriptional activity of PPARγ protein in adipose tissue [111,112]. O-GlcNAc is also involved in modulating the Wnt signaling pathway by negatively regulating β-catenin levels in the nucleus to decrease its transcriptional activity [113]. Even though β-catenin is O-GlcNAcylated on multiple sites, the competition between OGN and phosphorylation by GSK3β on the T41 residue is crucial for controlling β-catenin degradation. OGN stabilizes β-catenin in the cytosol [114] (Figure 2). Besides, GSK3β is a substrate for OGT, and OGN of GSK3β positively regulates its kinase activity in HEK-293FT cells [115]. The results indicate that O-GlcNAc contributes to adipogenesis progression. In combination, primary cilia and O-GlcNAc have controversial roles in adipocyte differentiation. Primary cilia-mediated Wnt signaling promotes β-catenin translocation into the nucleus and initiates transcription of adipogenesis repressor genes, thus curbing adipocyte differentiation. However, elevated OGN levels promote differentiation by excluding β-catenin from the nucleus, enabling the transcription of adipogenesis regulators. O-GlcNAc also activates GSK3β to stabilize PPARγ and promote adipocyte differentiation. 

Therefore, I propose that primary cilia and O-GlcNAc regulated signaling pathways happen sequentially during adipocyte differentiation. Ciliary signaling pathways, including Wnt, HH, and RPGRIP1L, function first to maintain the naïve preadipocyte status. Subsequently, elevated O-GlcNAc levels hamper ciliary signaling and facilitate adipocyte differentiation. The proposal is supported by observations of dampened adipocyte differentiating rates and shortened primary cilia length of adipose-derived mesenchymal stem cells (ASCs) in obese individuals. Restoration of primary cilia in obese ASCs improves their differentiation capacity [36,116]. Because O-GlcNAc is a negative regulator of primary cilia, elevated cellular OGN levels may cause defects in primary cilia and adipocyte differentiation in obese individuals.

### 2.3. Primary Cilia Regulate Glucose Homeostasis through Function in Pancreatic Development and Insulin Secretion

#### 2.3.1. Primary Cilia Regulate Pancreatic Development

The development of endocrine and exocrine pancreases is derived from the endodermal progenitors under the activation of a hierarchy of transcription factors, such as *pancreatic and duodenal homeobox factor 1 (Pdx1)*, *Neurogenin3 (Ngn3)*, and the downstream regulator *Neuronal differentiation 1 (NeuroD1)* [117]. *Pdx1*, an efficient early pancreatic progenitor marker, is critical in inducing the differentiation of endocrine and exocrine cells. The expression of *Pdx1* is mostly restricted to β-cells to induce insulin gene transcription after pancreatic development [118,119]. The induction of *Ngn3* expression in PDX1 positive progenitor cells is important for pancreatic endocrine cell differentiation [118,119]. The *Ngn3* null mutant mice showed no pancreatic endocrine cells [120]. The NeuroD regulates beta-cell development and the expression of the insulin gene; its deficiency strongly disturbs pancreatic islet development in mice. Primary cilia-mediated sonic hedgehog (Shh) and Wnt signaling pathways are involved in pancreatic development through integration of pancreatic development transcription factors [39]. Both *Pdx1* and *pancreas specific transcription factor-1a (Ptf1a)* play vital roles in the specification of multipotent progenitor cells’ (MPCs) lineage and the differentiation of pancreatic endocrine and exocrine cells. *Pdx1* or *Ptf1a* null mutant mice both showed pancreatic development defects in forming acinar cells, which are the functional unit of the exocrine cells [118,121]. The expression of Shh is restricted to the pancreatic bud endoderm to enable the high expression of Pdx1, which indicates the negative regulation to the Pdx1 expression [121,122,123]. Deficiency of *Pdx1* caused by ectopic expression of Shh in the pancreatic epithelium results in the mis-localization of pancreatic endocrine and exocrine pancreatic cells into gut mesoderm [123]. Loss of Shh results in increased pancreatic endocrine cell numbers and enlarged pancreas in mice [122]. Moreover, the repression of primary cilia-mediated Wnt/β-catenin signaling is essential for the differentiation of endocrine and exocrine cell types and pancreatic development [124]. Any ectopic expression *Wnt* under the control of Pdx1 promoter or β-catenin results in the defection of exocrine or endocrine pancreatic tissues. The results are consistent with the studies showing that elevating Wnt signaling through knockdown of β-catenin inhibitor *Apc* ceased pancreas development [125,126]. However, Wnt/β-catenin signaling is essential for the proliferation of pancreatic progenitors later on [127,128]. In combination, the tight regulation of the activity of transcription factors mediated by primary cilia is critical for proper pancreatic differentiation.

#### 2.3.2. Primary Cilia Regulate Glucose Homeostasis and Insulin Secretion

The implication of β-cell primary cilia function in maintaining glucose homeostasis and insulin secretion is suggested by the β-cell-specific inducible cilia knockout (βICKO) mice. The mutant mice showed significant impairment of glucose handling 4 weeks post-induction of primary cilia ablation followed by impaired glucose tolerance after 12 weeks when comparing to the vesicle-treated controls [129]. The loss of β-cell primary cilia decreases β-cells numbers and attenuates insulin secretion over time. The hyperphosphorylated ephrin-type A receptors 2 and 3 (EphA2/3), regulators of insulin secretion, attenuate insulin secretion by activating downstream extracellular signal-regulated MAPkinase/mitogen-activated-protein kinase (ERK/MAPK) in the βICKO islets [129,130]. The results are confirmed by the Ins1-Cre β-cell cilia knockout (βCKO) mouse, which showed decreased insulin secretion, glucose intolerance, and diabetes development. Moreover, β-cell cilia mediate the crosstalk with α- and δ-cells in the islets to regulate nutrient metabolism. Therefore, β-cell cilia deficiency attenuates α-cell and δ-cell hormone secretion and reduces glucose response [131]. Disrupting the integrity of basal body/ciliary in pancreas islet by knocking down basal body gene *Oral-facial-digital syndrome I (Ofd1)* or *Bbs4* dampens insulin secretion due to the disruption of the microtubular (MT) network and MT transport, such as exocytosis [37,132]. Insulin from β-cells is released from large dense core vesicles (DCVs) via exocytosis [133]. Docking of the vesicle to the plasma membrane is mediated by the pairing of vesicle membrane SNARE (soluble N-ethylmaleimide-sensitive factor attachment protein receptor) proteins, the v-SNAREs, and target membrane SNAREs, the t-SNAREs [133]. The attachment of SNARE subunit Snap25 on the target membrane and Syntaxin1 A (Stx1a) on the β-cell is required for the first phase of insulin release [134,135] (Figure 3). In the *Bbs4* or *Ofd1*-depleted pancreas islets, Snap25 and Stx1a protein levels are significantly decreased due to the loss of PI3K/FoxO1, which further result in the attenuation of exocytosis and insulin secretion [135]. Moreover, IRs localize on the primary cilium of insulin-stimulated glucose-responsive MIN6m9 mouse cells and human primary β-cells in response to insulin stimulus. In disrupted basal body/cilia mutants, IR trafficking to primary cilia is lost [37]. Deletion of *Ofd1* or *Bbs4* in MIN6m9 cells diminished hosphor-Akt (pSer473) upon insulin stimulation, indicating that primary cilia are necessary for transducing insulin signals [37]. These results highly suggest the primary cilia of the β-cells as a regulator of insulin secretion, insulin signaling, and glucose homeostasis.

Basal body/ciliary defection contributes to pre-diabetic phenotypes. Conversely, the diabetic Goto-Kakizaki (GK) rat showed ciliary dysfunction [37,136]. The phenocopies of GK rats are characterized by glucose intolerance and impaired insulin secretion. Even though β-cell density and endocrine cell relative volume showed normal readings, the pancreas of GK rats exhibited dramatically reduced ciliated β-cells compared to Wistar controls [37,136]. Molecular mechanical studies revealed that basal body/ciliary proteins, such as BBS4, IFT88, and KIF3a, were all upregulated. The upregulation of ciliary proteins is considered a compensatory mechanism for ciliary defection. Besides, the exocytotic protein level of Stx1a is downregulated [37,136,137]. Observation of primary cilia dysfunction and exocytosis impairment showed that they may contribute to T2DM susceptibility. A recent study reveals the importance of primary cilia in regulating islet function and preventing T2DM risk by comparing the transcriptome difference in pancreatic islets between the diabetes-prone New Zealand Obese (NZO) and diabetes-resistant B6-ob/ob mice. Among the 1048 genes expressed differentially in pancreatic islets of NZO and B6-ob/ob mice, 327 are annotated cilia genes. Furthermore, 81 of these cilia genes were detected in the RNA-seq-based expression analysis of the ciliary genes ortholog to humans in islets of diabetic donors. Compared to nondiabetic human donors, 64 cilia genes are downregulated, and only 17 genes are upregulated in diabetic human donors [38]. These results indicate that dysregulation of ciliary genes contributes to T2DM development. Indeed, the ciliation rate decreased more than 3-fold in NZO islets when compared to B6-ob/ob islets. In addition, NZO islets show less flexible regulation in blood with elevated glucose concentrations. In vitro study revealed that knockdown of KIF3A of ciliary protein expression in MIN6 cells attenuates ciliogenesis and reduces islet cell proliferation, which is one of the potential mechanisms of pancreatic cilia uses to regulate of energy homeostasis [38].

#### 2.3.3. O-GlcNAc Is Critical for the Pancreatic Development

O-GlcNAc’s involvement in the pancreas ranges from pancreatic development to insulin secretion. First, in the in vitro rat embryonic pancreatic development model, pharmacological inhibition of the HBP pathway decreased acinar cell differentiation and β-cell development accompanied by attenuated *Ngn3*, *Ptf1a,* and *Insulin* mRNA expression levels [117,119]. Treatment with OGA inhibitor, PUGNAc, promotes β-cell development and induces *NeuroD* and *Insulin* expression [117]. A most recent study showed that *Ogt* null mutation in the pancreatic epithelium impacts the development of endocrine and exocrine pancreases. RNA-seq revealed that OGT targets, such as *Pdx1*, *Ptf1a*, and *p53*, are downregulated to induce apoptosis of pancreatic progenitors [138]. Besides, the translocation of β-cell-specific transcription factor NeuroD1 from the cytosol to the nucleus under high glucose conditions is catalyzed by OGN to induce insulin expression in MIN6 cells [139]. Collectively, the O-GlcNAc modification is essential for activation or expression of key transcription regulators in pancreatic differentiation, including but not limited to *Pdx1, Ptf1a, and NeuroD*. Considering that cilia-mediated Shh and Wnt/β-catenin signals tightly monitor the activity or expression of the transcription factors above, O-GlcNAc is proposed to be a regulatory factor involved in primary cilia-mediated signaling cascades in pancreatic development.

#### 2.3.4. O-GlcNAc Regulates Glucose Homeostasis and Insulin Secretion of Pancreas

The elevation of OGN levels involved in the deterioration of insulin secretion was first observed in the pancreas of diabetic Goto-Kakizaki (GK) rats [140]. The GK pancreas showed higher OGT activity and OGN in both the islets and exocrine cells. Meanwhile, in vitro study showed that the elevated OGN levels blunted MIN6 β cells to glucose-stimulated insulin secretion, a phenomenon which is highly associated with decreased PDX1 levels in the diabetic islets [140]. Even though there is no direct evidence showing that primary cilia-mediated β-cells insulin secretion pathway through EphA2/3 are O-GlcNAcylated, their downstream insulin secretion activator ERK/MAPK is an up-regulator of O-GlcNAc [141]. Moreover, the regulation of O-GlcNAc in exocytosis vesicle-mediated insulin secretion is well studied. First, FoxO1, as a regulator of exocytosis, has its activity regulated by O-GlcNAc in response to glucose, and OGN of FoxO1 is increased in both diabetic hepatic and pancreatic β cells [142,143]. Snapin is another protein that forms a complex with SNAP25 to promote vesicle fusion and insulin secretion in pancreatic β cells. The phosphorylation of snapin on S50 is essential for pre-assembly with SNAP25, exocytosis vesicle fusion, and insulin secretion [144] (Figure 3). However, snapin pS50 is occupied by OGN in diabetic mice, which attenuates formation of the complex with SNAP25 and dampens insulin secretion [145]. Besides, both SNAP25 levels and the expression of Syntaxin 1 are decreased to reduce insulin secretion in pancreatic β-cells of GK rats. Overall, these observations show that O-GlcNAc is involved in regulating insulin expression and exocytosis-mediated insulin secretion. Though not many mechanisms have revealed how O-GlcNAc regulates insulin secretion, the participation of O-GlcNAc discussed here is directly or indirectly involved in the primary cilia-mediated pancreatic insulin secretion. 

Finally, O-GlcNAc is intensively involved in cross-talking with insulin signaling activity, as discussed previously. Not only is the recruitment of OGT to the plasma membrane triggered by insulin stimulation, but the activity of various insulin signaling molecules, such as IR, IRS, and PDK1, are regulated by OGN. O-GlcNAc acts as an inhibitor to attenuate the insulin pathway [146]. The trafficking of IRs to the primary cilia points to a connection with O-GlcNAc. Chronic OGN elevation may result in desensitization to insulin, further leading to hyperglycemia and insulin resistance, which are pre-symptoms of T2DM [147,148]. Given these observations, the dysfunction of primary cilia in diabetic mice is potentially caused by elevated O-GlcNAc levels.

## 3. Discussion

This paper discusses how primary cilia function in energy homeostasis regulation through hypothalamic neurons, preadipocytes, and pancreatic β-cells, as well as the participation of O-GlcNAc in primary cilia-mediated satiety signaling. In addition, we propose the role of abnormal OGN levels in primary ciliary dysfunction-mediated obesity/diabetes pathologies. Although the discussion about the primary cilia’s role is limited to only a few cell types and tissues, it still exhibits its variety in functions. Depending on the cell type, distribution of signaling receptors on the ciliary membrane varies, such as the hormone receptors, somatostatin, melanin, and leptin, on the ARC neurons and differentiating receptors, Wnt and HH, on the preadipocytes. In the quiet cells that lacked further differentiation, primary cilia respond to extracellular stimuli and transduce signals to the intracellular compartment via the ciliary receptors to regulate energy homeostasis. However, in differentiating cells, such as preadipocyte and pancreatic progenitors, primary cilia function as a differentiation suppressor, which is potentially a mechanism to monitor the progression of differentiation. As discussed in the paper, disruption of primary cilia integrity will induce elevated differentiation capacity in preadipocytes or increased pancreatic endocrine cell numbers.

Current findings demonstrate that O-GlcNAc, as a nutrient sensor, is widely involved in primary cilia pathways, especially in the leptin and insulin signaling in the ARC neuron, and in the pancreatic insulin secretion and signaling. Moreover, investigations showed that elevated O-GlcNAc shortens ciliary length both in in vitro and in vivo experiments. The dysfunction of primary cilia is seen in both obesity and diabetes mutant mice, and features higher OGN levels. Hence O-GlcNAc not only participates in the ciliary signaling but also regulates the proper formation of primary cilia. These observations allude to the hypothesis that dysregulation of O-GlcNAc contributes to primary cilia defects and promotes the progression of obesity/diabetes. To rationalize the role of O-GlcNAc in the regulation of primary cilia involved in energy metabolism, the following questions should be investigated. 

First, whether decreased OGN levels can restore primary cilia integrity and insulin sensitivity in cases obesity/diabetes. Second, whether the fluctuation of OGN levels during adipogenesis correlates with primary cilia formation and disassembly. Lastly, how O-GlcNAc involves primary cilia-mediated insulin signaling to maintain glucose homeostasis in pancreatic β-cells. These questions are worth investigating since it will build the connection between two nutrient sensors, primary cilia and O-GlcNAc, in the regulation of energy homeostasis and how the dysregulation of primary cilia and O-GlcNAc results in the pathologies of obesity and diabetes. Moreover, the crosstalk between primary cilia and O-GlcNAc in metabolic regulation can be discussed in more aspects, such as in the skeleton muscles and kidney. The dysfunction of primary cilia and dysregulation of O-GlcNAc is characterized both in cancers and neurodegeneration diseases, which magnifies the significance of the research.

## Figures and Tables

**Figure 1 biomolecules-10-01504-f001:**
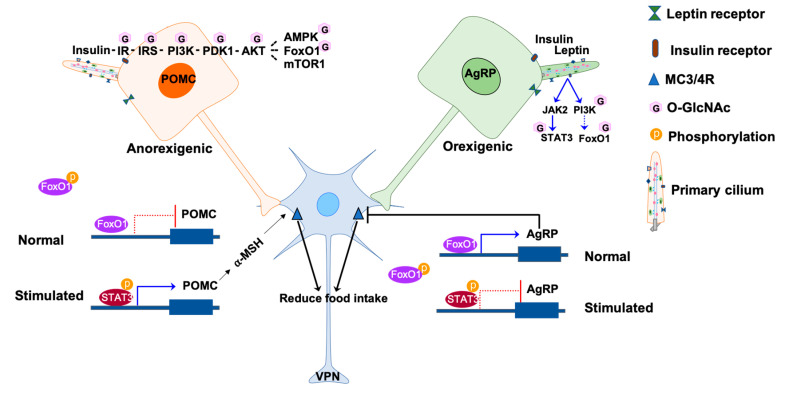
Involvement of O-GlcNAc in primary cilia signaling in the hypothalamic neurons. The activities of POMC and AgRP neurons are regulated by insulin and leptin hormones in response to nutrients. Under normal conditions, FoxO1 binds to the promoters *agrp* and *pomc* but have different effects. FoxO1 promotes the expression of AgRP to inhibit the activation of lateral neurons in the VPN through MC3/4R, while FoxO1 inhibits the expression of POMC. The binding of insulin or leptin to the receptors on the neuronal membrane phosphorylates and activates a downstream signal, such as JAK2-STAT3 and PI3K-PDK1-AKT-FoxO1. The phosphorylated FoxO1 is excluded from the nucleus, enabling the binding of phosphorylated STAT3 to the promoters. The binding of pSTAT3 to the *agrp* promoter inhibits AgRP expression thus releasing the inhibition to the lateral neurons. On the other hand, the binding of pSTAT3 to the *pomc* promoter promotes POMC expression and activate the lateral neurons to reduce hunger. The O-GlcNAcylated molecules are labeled with pink hexagons. The phosphorylated molecules are labeled with orange circles. IR, insulin receptors; IRS, insulin receptor substrate proteins; PI3K, phosphatidylinositol-3-kinase; PDK1, 3-phosphoinositide-dependent protein kinase 1; AKT, protein kinase B; AMPK, AMP-dependent kinase; FoxO1, forkhead box protein O1; mTOR1, mammalian target of rapamycin 1; JAK2, Janus kinase 2; STAT3, signal transducer and activator of transcription 3; MC3/4R, melanocortin 3 and 4 receptors; AgRP, agouti-related peptide; POMC, proopiomelanocortin.

**Figure 2 biomolecules-10-01504-f002:**
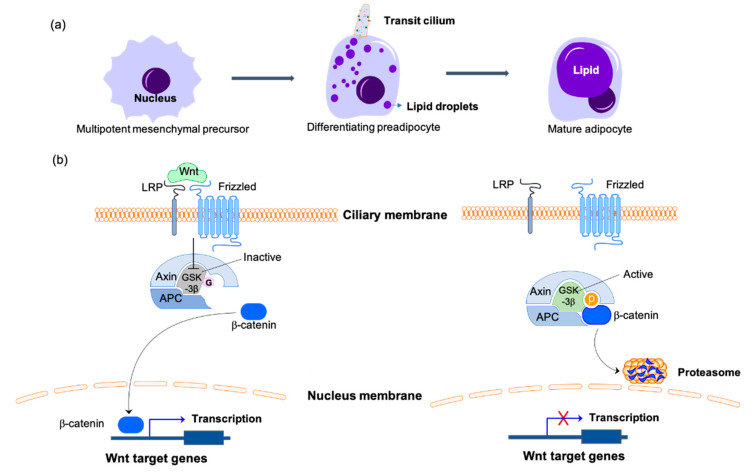
Primary cilia-mediated Wnt/β-catenin signaling in the regulation of adipogenesis. (**a**) Adipogenesis is the process of mesenchymal precursors committing to preadipocyte lineage cells then differentiating into mature adipocytes. Primary cilia are only present on the differentiating preadipocytes during this process. (**b**) On the differentiating preadipocytes, the binding of Wnt to the receptors of the ciliary membrane catalyzes the phosphorylation and inactivation of GSK3β, thus inhibiting the phosphorylation of β-catenin. β-catenin is transported to the nucleus to initiate the transcription of Wnt target genes to inhibit the differentiation of preadipocytes. With Wnt missing or the primary cilium disassembling, the active form of GSK3β phosphorylates β-catenin and promotes its degradation by the proteasome, which inhibits the expression of adipogenesis inhibitors and promotes the differentiation of preadipocytes. The O-GlcNAcylated molecules are labeled with pink hexagons. The phosphorylated molecules are labeled with orange circles. LRP, lipoprotein receptor-related protein; APC, adenomatous polyposis coli protein; GSK3β, glycogen synthase kinase-3β.

**Figure 3 biomolecules-10-01504-f003:**
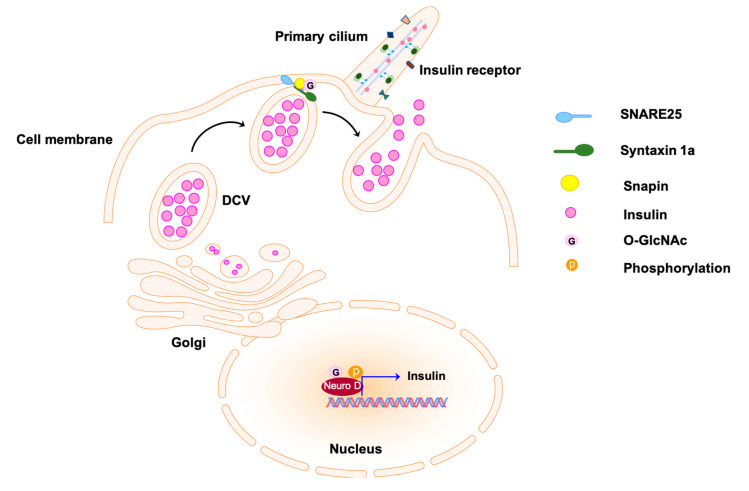
Insulin secretion from pancreatic β-cells by exocytosis. Transcription regulator Neuro D promotes the expression of insulin; it is enclosed in the dense core vesicles (DCVs) and secreted to the extracellular matrix via exocytosis. The docking of the exocytosis vesicles to the target membrane is mediated by the formation of complex SNARE25-Syntaxin1a at the first stage of insulin secretion. The O-GlcNAcylation of snapin, a subunit of the complex, is involved in the regulation of complex formation and vesicle fusion. The O-GlcNAcylated molecules are labeled with pink hexagons. SNARE25 indicates soluble N-ethylmaleimide-sensitive factor attachment protein receptor.

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
