# Peer review of "Potential Roles of O-GlcNAcylation in Primary Cilia- Mediated Energy Metabolism"

_biomolecules, 2020, doi:10.3390/biom10111504_

Round 1

Reviewer 1 Report

This is nicely written, the review gives a good overview over the topic. 

Minor points:

The abbreviations reported in the figures are missing in the legends of the figures, the authors should specify the abbreviations in the legends

Figure 1. The title of the legend misses “in hypothalamic neurons”

Figure 3. the insulin signaling is not clear what is referring to. What is the G inside the pink rhombus? And the P? they are reported in the figure 1 but not in the figure 3. Please insert these in the legend

The pink circles which are the insulin are missing in the symbols legend, the authors should add it

Paragraph 1.1.1; 1.1.2 and 1.1.3: the words “first, second and third” should be deleted from the titles

Reviewer 2 Report

In this review article, Prof. Tain present an elegant correlation of O-GlcNAc, primary cilia and energy homeostasis. The concept and organization of this manuscript are interesting. Besides, the manuscript is well written, and some minor suggestions are shown below.

  1. First, secondary, and thirdly were used int the subtitles of 1.1.1, 1.1.2, and 1.1.3, please remove these words as they are not necessary.

  1. Figure 2 should be subdivided into 2A, Differentiation of adipocyte, and 2B, Cilia-mediated Wnt signaling in adipogenesis. In addition, Nucleus and lipid droplet showed be indicated in figure 2A.

  1. Please show the solid pink circle as insulin in the icon in figure 3.
